# Arthroscopy-Assisted Corrective Osteotomy, Reduction, Internal Fixation and Strut Allograft Augmentation for Tibial Plateau Malunion or Nonunion

**DOI:** 10.3390/jcm9040973

**Published:** 2020-04-01

**Authors:** Jr-Yi Wang, Chun-Ying Cheng, Alvin Chao-Yu Chen, Yi-Sheng Chan

**Affiliations:** 1Department of Orthopedics, Shuang Ho Hospital, Taipei Medical University, Taipei 11031, Taiwan; active9102078@gmail.com; 2Department of Orthopedics, School of Medicine, College of Medicine, Taipei Medical University, Taipei 11031, Taiwan; 3Department of Orthopedic Surgery, Chang Gung Memorial Hospital and Chang Gung University College of Medicine, Taoyuan 333, Taiwan; orthhand@cgmh.org.tw (C.-Y.C.); alvin_ortho@yahoo.com (A.C.-Y.C.); 4Bone and Joint Research Center, Chang Gung Memorial Hospital, Taoyuan 333, Taiwan

**Keywords:** Arthroscopy-assisted corrective osteotomy, tibial plateau malunion, tibial plateau nonunion

## Abstract

Purpose: The purpose of this study was to present the results of arthroscopy-assisted corrective osteotomy (AACO), reduction, internal fixation, and strut allograft augmentation for tibial plateau malunion or nonunion. Methods: Fifty-eight patients, mean age 49 ± 11.9 years old, with tibial plateau malunion (*n* = 44) or nonunion (*n* = 14), were included in this study. There were 19 Schatzker type II fractures (32.7%), 2 type III fractures (3.4%), 7 type IV fractures (12%), 20 type V fractures (34.5%), and 10 type VI fractures (17.2%). The mean follow-up period was 46.2 ± 17.6 months. Clinical and radiologic outcomes were scored by the Rasmussen system. Articular depression was measured from computed tomography. Secondary osteoarthritis was diagnosed when radiographs showed a narrowed joint space in the injured knee at follow-up. Results: Mean clinical score improved from 15.4 ± 3.9 (pre-revision) to 23.2 ± 4.5 (post-revision). Mean radiologic score improved from 7.7 ± 2.5 (pre-revision) to 12.0 ± 3.9 (post-revision). Fifty-six fractures achieved successful union. The average union time was 19.6 ± 7.5 weeks. Post-revision, 81% had good or excellent clinical results and 62% had good or excellent radiological results. Secondary osteoarthritis (OA) was noted in 91% of all injured knees, where 25.8% were mild OA, 25.8 % were moderate OA, and 38% were severe OA. There were 6 cases of deep infection (10.3%) and 1 case of wound edge necrosis (1.7%). Five cases were converted to total knee replacement after the index surgery with an average period of 13.5 months (range 8–24 months). Conclusions: Arthroscopy-assisted corrective osteotomy, reduction, internal fixation, and strut allograft augmentation can restore tibial plateau malunion/nonunion with well-documented radiographic healing and good clinical outcomes.

## 1. Introduction

Management of tibial plateau fractures is challenging. Malreduction, improper fixation, neglected intra-articular lesions, and inadequate management of bone defects lead to the failure of surgical treatment [1,2,3,4,5]. Traditionally, open corrective osteotomy with bone grafting is the standard of care following malunion [2,4,5,6,7,8,9]. However, arthrotomy with wide soft tissue dissection may have the higher incidence of complications and longer recovery time for the patient [2,4,5,9].

The advantage of arthroscopically-assisted osteosynthesis of fractures of the tibial plateau are direct visualization of intra-articular fractures, accurate fracture reduction, reduced morbidity in comparison with arthrotomy, simplified diagnosis and treatment of meniscal and ligamentous injuries, thorough-joint lavage, and removal of loose fragments [10,11,12,13,14,15,16]. Good early to medium-term results of arthroscopically-assisted osteosynthesis of tibial plateau fractures have been reported previously [10,11,17,18,19,20]. The purpose of this study was to present the radiological and clinical results of tibial plateau malunion or nonunion treated with arthroscopically-assisted corrective osteotomy (AACO) surgery. To our knowledge, this unique and novel method is not reported in the literature. We hypothesized that AACO surgery for tibial plateau malunion or nonunion wound allow accurate, stable alignment and articular surface reduction with minimal dissection and a satisfactory outcome.

## 2. Materials and Methods

From January 2002 to July 2013, 63 consecutive patients with tibial plateau fracture malunion or nonunion underwent arthroscopically-assisted corrective osteotomy (AACO). Fifty-eight patients (92%) who were followed up for over 24 months were enrolled in this retrospective case series. All patients were given a consent form before surgery and underwent the same treatment protocol for arthroscopy-assisted corrective osteotomy, reduction, internal fixation, and strut allograft augmentation for tibial plateau malunion or nonunion. The study was conducted in accordance with the Declaration of Helsinki, and the protocol was approved by the Ethics Committee of Chang Gung Memorial Hospital (Project identification code 104-8592B). The mean age at operation was 49 ± 11.9 years old (range: 22 to 75 years). The follow-up period was 46.2 ± 17.6 months. No patients were lost to follow-up in this consecutive case series. Tibial plateau fractures were categorized by the Schatzker classification according to initial fracture patterns [21]. Table 1 lists patient demographics. Indications for operative fixation included any varus instability exceeding 10° of medial tibial plateau fracture at full extension or lateral plateau fracture with valgus instability exceeding 10° and articular step-off exceeding 3 mm or tibial condylar widening exceeding 5 mm. Patients were excluded if they had any of the following features: active or previous infection in the affected knee joint, poor soft-tissue status, knee ankylosis, bad compliance, morbid obesity, severe systemic illness (diabetic mellitus under poor control, liver cirrhosis Child B/C, active cancer, chemotherapy, hemophilia, or a medical contraindication for surgery). Those who followed up for more than 24 months were enrolled in this study.

Preoperative evaluations included detailed physical examination of the condition of the soft-tissue envelope, sensory-motor function of the limb, and vascular status in terms of pulsation over the dorsalis pedis and posterior tibialis arteries. All patients underwent plain film study in anteroposterior and lateral views as well as computed tomography of each knee. The mean duration between the first time surgery to revision surgery was 9.2 ± 5.7 months. All preoperative evaluations and surgical procedures were performed by a single surgeon (Y-S.C.).

### 2.1. Surgical Technique

The patients were positioned supine on the operating table and given general endotracheal anesthesia. Before surgery, a complete knee evaluation was performed with the patient under anesthesia. A pneumatic tourniquet was applied to the thigh, but a leg holder was not required because most arthroscopic manipulation was conducted in the “figure 4” position. The anterolateral and anteromedial portals were used to insert the arthroscope as well as working instruments. The arthroscopic joint inspection permitted evaluation of the cartilage condition, debridement of fibrous tissue, and localization of fracture malunion site (Figure 1A). The cartilage injury grading was recorded as Outerbridge classification [22]. The capsuloligamentous structures were then probed and the associated intra-articular lesions were evaluated. Incisions were made on the side of the fracture directly medial-lateral (types I through IV) and bilaterally for types V and VI, starting from approximately 1 cm proximal to the articular surface and extending approximately 8 cm distally. The previous plate and screw were removed and wound culture was done routinely. The tibial metaphysis was carefully exposed and care was taken to avoid arthrotomy and minimize periosteal stripping. The depressed fragment or malunited area was located with a standard tibial guide for anterior cruciate ligament (ACL) reconstruction and a Kirschner wire was inserted through the metaphysis into the edge of malunited fracture line (Figure 1B). Another Kirschner wire repeated the aforementioned procedure. Then, a 0.5 cm osteotome was started at the metaphyseal area to the articular area on the plane that the two Kirschner wires created to completely separate the bone malunion area (Figure 1C). The method was repeated several times until the depressed articular fragment was freely elevated by bone tamp (Figure 1E) and using Kirschner wire to temporarily fix the elevated fragment. The articular height was checked under fluoroscope exam and arthroscopically.

The metaphyseal area may have revealed a huge bone defect. We measured the length, width, and depth of the defect area. Then, the fresh frozen femoral head allograft were defrosted and cut by oscillator saw to fit the previous measured bone defect. The overlying cartilage were removed as cleanly as possible. Then, this cancellous strut allograft was impacted into the metaphyseal bone defect area (Figure 1F). The residual small defect was filled with artificial bone graft and GeneX (Biocomposites Ltd., Staffordshire, UK) putty (β-tricalcium phosphate/calcium sulfate hemihydrate compound) mixed with vancomycin powder. Internal fixation was applied with either a nonlocking L buttress plate (Depuy Synthes 4.5 mm L buttress plate) or a locking plate (Depuy Synthes LCP Proximal Tibial Plate 3.5) and interfragmentary screw (Depuy Synthes Standard screws 3.5 mm, cortical or cancellous) fixation. Intraoperative radiographs were routinely taken for all types of fractures to reconfirm adequate reduction before wound closure.

The associated intra-articular pathology was treated in the appropriate sequence after fracture fixation. Meniscal tears were repaired when they were within 5 mm of the meniscosynovial junction. Degenerative meniscus tears were debrided and trimmed to stable end. ACL injuries were treated by arthroscopy-assisted fixation of the ACL avulsion fracture in a 1-stage surgical procedure and second-stage reconstruction after fracture healing for complete ACL tears.

### 2.2. Clinical and Radiologic Assessment

The pre-revision and post-revision clinical and radiologic scales were recorded according to the Rasmussen system (Table 2 and Table 3) [23]. The post-revision scales was recorded at the latest followed up outpatient department visit. The Rasmussen system was intended to improve upon the Hohl and Luck system [24] by making it more quantitative. The advantage of the Rasmussen system is its analysis of both functional and anatomic end results for tibial plateau fractures after treatment. This scoring system is widely used in related studies of tibial plateau fractures [11,25,26,27,28]. The follow-up protocol included analysis of subjective complaints and objective clinical findings. All 58 patients completed a questionnaire regarding their overall function. Subjective data were collected to assess swelling, difficulty climbing stairs, joint stability, ability to work and participate in sports, and overall patient satisfaction with recovery. Range of motion (ROM) of the knee joint was measured with the patient in a prone position with test-side ankle off plinth. A goniometer axis was placed at the skin prominence of lateral epicondyle of the femur. A stationary arm was placed along the femur to greater trochanter. A movement arm was placed along the fibula to lateral malleolus. Measurement of ROM was documented at each OPD visit. We only presented the preoperative ROM and most recent postoperative follow-up ROM. All patients were evaluated by the senior author (Y-S.C.).

A radiographic evaluation, including bipedal examination and examination of both knees, was done preoperatively, at 3 months, 6 months, and 1 year postoperatively, and then annually thereafter. Long-leg standing films (with standing scanography used to measure the tibio-femoral angle) were obtained annually postoperatively.

In each case, preoperative and postoperative articular depression were measured from the Computed tomography (CT) coronal series. The amount of articular depression was measured from the opposite remaining articular surface. A line was drawn at the level of the normal articular surface and extended across the depressed area. A measurement was made from this line to the point of maximum depression. When both plateaus were involved, a line was drawn at the level of the femoral condyles and another line parallel to it was drawn through the base of the tibial spine, then lines were drawn to the point of maximum depression of both plateaus. Magnetic resonance imaging scans were used to identify meniscal disorders and ligamentous injuries.

Union was defined as the presence of a bridging callus on 2 radiographic views [29]. Nonunion was defined as a failure of progressive radiographic healing over a 1 year period [2]. Osteoarthritic changes were judged according to Ahlbäck’s scale [30] and the mechanical axis was measured by radiography. On the basis of Ahlbäck’s scale, osteoarthritis was estimated by narrowing of the joint space. Secondary osteoarthritis was defined as follows: if the radiograph showed a narrowed joint space in the injured knee at follow-up, the comparison was with the film taken at the time of injury. If the findings were comparable, no secondary degenerative change was recorded. If relative narrowing was noted, comparison was then made with the film of the uninjured knee. Relative narrowing of the joint space by less than 50% was considered mild, relative narrowing by greater than 50% was considered moderate, and 100% (i.e., obliterated) narrowing was considered severe [31].

### 2.3. Statistical Methods

Each radiograph was evaluated by 5 observers (visiting orthopedic staff) in a blinded fashion to assess interobserver variance. The 5 observers also classified fracture type and measured articular depression in the 58 patients. Statistical analysis determined the interobserver variance in the diagnosis for secondary osteoarthritis, fracture type, and articular depression. The interobserver variability by use of the kappa method in the diagnosis of secondary osteoarthritis, fracture type, and articular depression was in all cases insignificant (*p* < 0.05). The Mann–Whitney U test was used for comparing preoperative and postoperative clinical and radiological outcomes. *P* < 0.05 was recognized as significance.

## 3. Results

Data of the surgical patients are shown in Table 1. There was no significant relation between fracture type and patient age, gender, or injured limb (*p* > 0.05). The average range of motion was 4.4 ± 9.18 to 98.0 ± 28.5 preoperatively and 1.5 ± 4.0 to 112.7 ± 18.8 postoperatively. The tibial-femoral angles were 7.3 ± 3.3 (type II), 8.0 ± 2.8 (type III), −2.4 ± 5.44 (type IV), −1.5 ± 9.6 (type V), and 0.4 ± 7.6 (type VI) preoperatively and were corrected to 6.7 ± 5.7 (type II), 6.0 ± 0.0 (type III), 4.6 ± 4.3 (type IV), 3.4 ± 9.3 (type V), and 1.0 ± 7.5 (type VI) postoperatively. The preoperative depression of the plateaus were 15.9 ± 6.2 (type II), 7.5 ± 3.5 (type III), 16.4 ± 3.8 (type IV), 19.5 ± 7.7 (type V), and 20.7 ± 5.6 (type VI) and were reduced to 2.8 ± 3.6 (type II), 0.0 ± 0.0 (type III), 2.7 ± 4.0 (type IV), 5.1 ± 4.9 (type V), and 5.5 ± 3.4 (type VI) postoperatively. In total, the amount of depression of the plateaus were 17.7 ± 6.8 preoperatively and 3.9 ± 4.2 postoperatively. Significant difference was noted in type II, IV, V, VI, and overall (Figure 2 and Figure 3).

### 3.1. Clinical Assessment

Table 4 shows preoperative and postoperative clinical assessment results. Preoperatively, the clinical Rasmussen scores were 17.1 ± 1.9 (type II), 20.0 ± 4.2 (type III), 15.6 ± 4.4 (type IV), 14.9 ± 4.2 (type V), and 12.1 ± 4.1 (type VI). Postoperatively, the clinical scores increased to 22.6 ± 5.3 (type II), 26.0 ± 0.0 (type III), 25.3 ± 1.4 (type IV), 23.6 ± 4.0 (type V), and 21.6 ± 5.2 (type VI). In total, the clinical scores were 15.4 ± 3.9 preoperatively and 23.2 ± 4.5 postoperatively. Significant difference was noted in type II, IV, V, VI, and overall. The clinical satisfactory rate improved from 10.3% preoperatively to 81% postoperatively.

### 3.2. Radiologic Assessment

All 58 fractures, except 2, achieved successful union (these 2 nonunion cases shifted to total knee arthroplasty). The average union time was 19.6 weeks. Table 5 shows preoperative and postoperative radiologic assessment results. Preoperatively, the radiological Rasmussen scores were 8.9 ± 1.4 (type II), 13.0 ± 1.4 (type III), 8.0 ± 2.3 (type IV), 6.6 ± 2.6 (type V), and 6.2 ± 2.0 (type VI). Postoperatively, the radiological scores increased to 13.3 ± 3.2 (type II), 17.0 ± 1.4 (type III), 14.3 ± 3.3 (type IV), 10.7 ± 4.2 (type V), and 9.6 ± 2.6 (type VI). In total, the radiological scores were 7.7 ± 2.5 preoperatively and 12.0 ± 3.9 postoperatively. Significant difference was noted in type II, IV, V, VI, and overall. The radiological satisfactory rate improved from 8.6% preoperatively to 62% postoperatively.

### 3.3. Associated Injuries and Procedures

Of the 58 patients in this series, 31 (53.4%) had associated intra-articular lesions (meniscus, cruciate ligament, collateral ligament) (Table 1). Among these 31 patients, the meniscal injury was noted in 25 knees. There were 7 medial and 21 lateral meniscal injuries. Three knees had both medial and lateral meniscal pathology. Five menisci were sutured, 23 were partially resected, and none were totally removed. Ligament injuries were noted in 11 knees with 17 lesions, including 4 ACL complete ruptures, 4 ACL partial ruptures, 1 PCL complete rupture, 3 PCL partial ruptures, and 1 medial collateral ligament partial rupture. There were 4 combined ligament injuries (ACL partial tear and PCL partial tear in three knees and ACL complete tear and PCL complete tear) in one knee. High-grade chondral injuries (Outerbridge Gr 3, Gr 4) were noted in 11 type II, 2 type IV, 9 type V, and 6 type VI fractures. In total, 28 (48.3%) knees had high-grade chondral injuries.

### 3.4. Complications

Secondary osteoarthritis was noted in 53 injured knees (91%). The relative narrowing of the joint space was mild in 15 knees (25.8%; 7 type II, 1 type IV, 4 type V, and 3 type VI), moderate in 3 knees (25.8 %; 4 type II, 4 type IV, 4 type V, and 3 type VI), and severe in 23 knees (38%; 6 type II, 1 type III, 2 type IV, 10 type V, and 4 type VI). In the 58 patients, deep infection was noted in 6 cases (10.3%). Wound debridement, sequestrum removal, and copious irrigation were done and parenteral antibiotics were used. The implants were removed if the screws were loosening and left in place if the screw and plate were stable. In the six cases of deep infection, five cases were cured without recurrent infection after one to two debridement surgeries. Two cases develop knee arthrofibrosis, with the range of motion only 0 to 30 degree. Arthroscopic release was done for these two patients at the same time when removing the locking plate and screws.

There was one case of wound edge necrosis that was noted 1 month after the index surgery. Rotation musculocutaneous flap was done for wound coverage. The flap and wound healed well. The osteotomy site healed well too.

Five cases (3 type II, 1 type V and 1 type VI) needed to shift to knee arthroplasty due to persistent pain, nonunion, and radiograph-confirmed severe post-traumatic osteoarthritis. The average interval between the index surgery to total knee arthroplasty was 13.5 months (range 8–24 months). No complications directly associated with arthroscopy were noted in any of the 58 patients.

## 4. Discussion

In this report, we presented the surgical technique and radiographic and clinical outcomes of arthroscopic-assisted corrective osteotomy in the treatment of tibial plateau fracture malunion or nonunion. Postoperatively, 81% patients achieved satisfactory clinical results and 61% patients achieved satisfactory radiographic results. Significant improvement was confirmed by comparing preoperative and postoperative conditions.

Complication after tibial plateau fracture surgery has been reported by several authors [1,3,32]. Papagelopoulos et al. divided the complications into early (i.e., loss of reduction, deep vein thrombosis, infection) or late (i.e., malunion and nonunion, implant breakage, post-traumatic arthritis) categories [1]. Huang et al. analyzed the causes of failure for their 25 patients who had failed tibia plateau surgery and revealed that 19 (76%) had inadequate fixation, 21 (84%) had malreduction, and 25 (100%) had bone defects [3].

Knee arthroplasty for post traumatic knee arthritis following tibial plateau fracture malunion or nonunion has been reported by many authors. However, according to different series, higher complication rates: 13.7%–34%, poor outcomes: 19%–23%, soft-tissue compromise: 5%–13%, stiff knee: 4%–11.5%, high revision rate 4%–11%, and deep infection: 3%–5% were noted, where all of the above were higher than primary knee arthroplasty for degenerative knee osteoarthritis [33,34,35,36,37,38,39,40]. Besides, 12%–13% of patients need semi-constrained or hinge knee implants for compensation of the post-traumatic articular depression and ligament imbalance.

Corrective osteotomy could maintain the native knee joint, restore lower extremity alignment, preserve the bone stock, delay joint replacement, and create better conditions for an eventual replacement. These reconstructive surgical strategies were mentioned in only a few studies [2,4,9]. Wu et al. presented their technique for correction of the tibia plateau fracture malunion with varus deformity by tibia shaft osteotomy distal to the tibial tubercle, realigning the lower extremity axis, and medial fixation by 95° angled blade plate [2]. Postoperatively, the proximal medial tibial angle (PMTA) was corrected to the acceptable range (80°–99°) in all patients and knee function improved in 88% patients. However, with this technique, intraarticular pathology, such as intraarticular step-off and associate intraarticular soft tissue pathology, cannot be managed properly. Kerkhoffs et al. reported their technique with combined intra-articular and varus opening wedge osteotomy for lateral depression and valgus malunion of the tibia plateau [4,5]. According to their technique, with a standard lateral arthrotomy, the anterior 50% to 60% of the lateral plateau can easily be visualized to expose more posteriorly situated depressions, an osteotomy of the Gerdy tubercle and reflection of the attached iliotibial tract allow visualization of approximately 80% of the lateral plateau, and an additional osteotomy of the fibular head allows full anterior dislocation of the lateral tibial plateau. An osteotomy of the tibial tuberosity is necessary when there is a combination of medial and lateral malunions. Although the authors reported that excellent results were achieved in 74% patients, possible disadvantages of extended arthrotomy and delayed healing of tibial tubercle osteotomy site may be encountered.

Recently, computer-assisted planning and patient-specific surgical guidelines have been proposed to correct tibial plateau post-traumatic malunion [41,42,43]. Initial results are encouraging. However, this technique must mobilize the soft tissue attachment to fully position the guide on the bone. This includes the pes anserinus and the ilio-tibial band. In addition, additional CTs will be required if the opposite side is used as a reconstruction template, resulting in increased radiation exposure.

We presented the surgical technique and radiographic and clinical outcomes of arthroscopic-assisted corrective osteotomy in the treatment of tibial plateau fracture malunion or nonunion. In our series, postoperatively, 81% patients achieved satisfactory clinical result and 61% patients achieved satisfactory radiographic result. Significant improvement was confirmed by comparing preoperative and postoperative conditions. This technique can be used in all type of tibial plateau malunion and nonunion fractures. Of the 58 patients, 51.7% suffered Schatzker type V or type VI fractures and could be managed by the technique very well with significant improvement both clinically and radiologically. However, according to the Rasmussen score of radiological results, fractures with type V or type VI would get relatively lower scores. This may explain why our series had only a 61% for satisfactory radiologic results.

By the assistance of arthroscopy, the fracture malunion site could be identified, the corrective osteotomy were created, and the depressed intraarticular fragment could be elevated under direct visualization. Besides, the diagnosis, documentation, and management of associate intraarticular soft tissue pathology could be performed smoothly under arthroscopy at one-stage surgery. Our technique avoided arthrotomy and fibulotomy. To the best of our knowledge, this is the first case series presenting the new technique.

In our series, we had five cases that needed to convert to total knee arthroplasty due to persistent pain, nonunion, and radiograph-confirmed severe post-traumatic osteoarthritis with a mean conversion time of 13.5 months (range: 8–24 months). Three cases used primary total knee replacement and two cases used constrained condylar knee prosthesis (LCCK; Zimmer) and there was no need for rotating hinge type revision total knee replacement. There was no further complication related to the arthroplasty procedure.

However, some limitations and weaknesses of this study must be acknowledged: there was no control group, this was not a randomized study, only a single surgeon was involved, all patients were rated by the same author, the sample size was limited, and follow-up to determine further osteoarthritic change needs to be longer.

## 5. Conclusions

Arthroscopy-assisted corrective osteotomy, reduction, internal fixation, and strut allograft augmentation is a unique and new method to restore tibial plateau malunion/nonunion with well-documented radiographic healing, good clinical outcomes, and low complication rates.

## Figures and Tables

**Figure 1 jcm-09-00973-f001:**
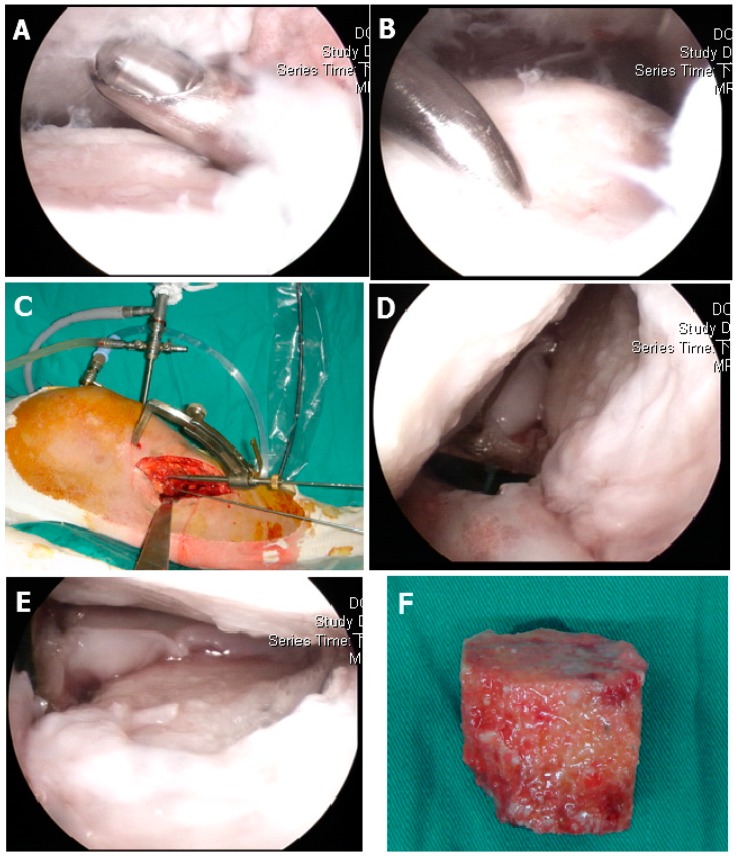
Operative procedure: (**A**) Arthroscopic documentation and debridement. (**B**) Use of a microfracture awl to identify the intra-articular depression zone. (**C**) Anterior cruciate ligament (ACL) tibial guiding for osteotomy axis and location, where two parallel Kirschner wire determined one plane, the 5 mm osteotome started from the metaphysic area to the articular surface. (**D**) The malunited fracture area was completely released. (**E**) The depressed articular fragments were elevated with bone tamp under direct arthroscopically inspection. (**F**) Bone grafting of the metaphyseal defect with structural allograft femoral head.

**Figure 2 jcm-09-00973-f002:**
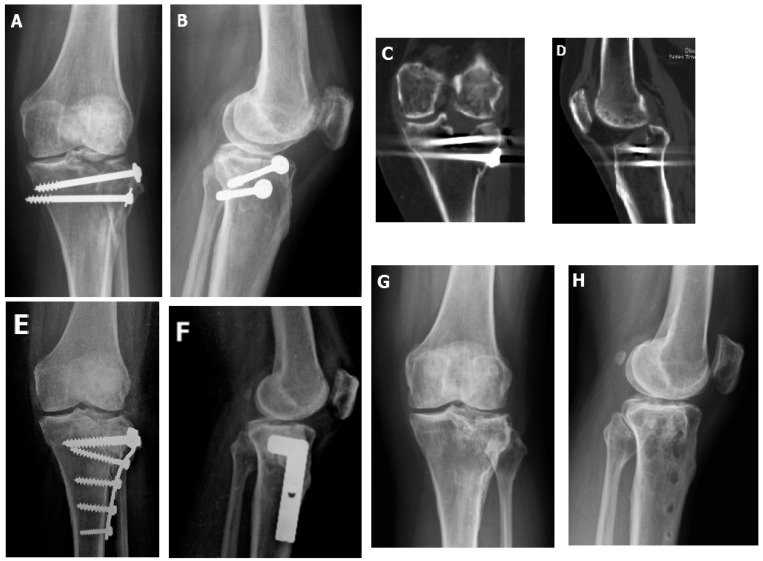
(**A**,**B**) This 43-year-old patient had a Type II fracture, where open reduction with internal fixation was done by two cannulated screws on the day of injury. This radiograph was checked four months after the initial surgery, where a lateral split with depression was noted. (**C**,**D**) CT revealed lateral plateau huge intraarticular split and depression. (**E**,**F**) Immediate postoperative radiograph after arthroscopically-assisted corrective osteotomy (AACO), bone grafting with structural allograft femoral head, and internal fixation by lateral plate. (**G**,**H**) 4 years after AACO, the plate was removed and the radiological score were excellent.

**Figure 3 jcm-09-00973-f003:**
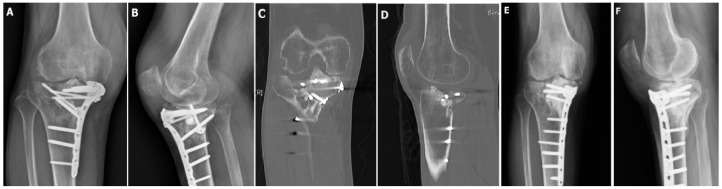
(**A**,**B**) This 57-year-old patient had a Type II fracture, where open reduction with internal fixation was done by medial locking plate 3 days after injury. When this radiograph was checked 14 months after the initial surgery, lateral plateau depression and widening were noted. (**C**,**D**) CT revealed lateral plateau huge intraarticular depression. (**E**,**F**) 2.5 years after AACO, the radiological score was excellent.

**Table 1 jcm-09-00973-t001:** Demographics of surgical patients: Schatzker classification of fracture types and associated soft-tissue injuries.

	Type II	Type III	Type IV	Type V	Type VI	Total
No. of patients (%)	19 (32.7%)	2 (3.4%)	7 (12%)	20 (34.5%)	10 (17.2%)	58 (100%)
Mean age (yr)	49.8 ± 12.9	34 ± 16.9	49.3 ± 7.2	49.5 ± 13.4	49.2 ± 8.3	49 ± 11.9
Gender (M/F)	5/14	0/2	1/6	6/14	4/6	16/42
Side (R/L)	7/12	0/2	0/7	9/11	7/3	23/35
**Preoperative condition**						
Range of motion	1.3 ± 3.3~102.3 ± 30.6	0.0 ± 0.0~110.0 ± 28.3	3.6 ± 6.3~115.0 ± 15.0	4.0 ± 7.7~93.7 ± 27.9	12.5 ± 16.2~84.0 ± 28.6	4.4 ± 9.18~98.0 ± 28.5
Mean depression plateau (mm)	15.9 ± 6.2	7.5 ± 3.53	16.4 ± 3.77	19.5 ± 7.7	20.7 ± 5.6	17.7 ± 6.8
Mean tibiofemoral angle	7.3 ± 3.3	8.0 ± 2.8	−2.4 ± 5.44	−1.5 ± 9.6	0.4 ± 7.6	-
**Postoperative condition**						
Range of motion (latest follow up)	1.1 ± 3.1~111.6 ± 22.4	0.0 ± 0.0~130.0 ± 0.0	0.0 ± 0.0~122.8 ± 7.6	1.3 ± 3.9~112.3 ± 17.0	4.5 ± 6.0~105.0 ± 19.0	1.5 ± 4.0~112.7 ± 18.8
Mean depression plateau (mm)	2.8 ± 3.6	0.0 ± 0.0	2.7 ± 4.0	5.1 ± 4.9	5.5 ± 3.4	3.9 ± 4.2
Mean tibiofemoral angle	6.7 ± 5.7	6.0 ± 0.0	4.6 ± 4.3	3.4 ± 9.3	1.0 ± 7.5	-
**Associated injuries**						
High grade chondral injury (Outerbridge Gr 3, Gr 4) (%)	11(57.9%)	0	2 (28.6%)	9 (45%)	6 (60%)	28 (48.3%)
Meniscus (%)	10	1	3	8	4	26 (44.8%)
ACL (%)	1	0	1	4	2	8 (13.8%)
PCL (%)	1	0	1	3	0	5 (8.6%)
MCL (%)	0	0	0	1	0	1 (1.7%)
Frequency of patients involved (%)	52.6% (10/19)	50%	57.1% (4/7)	55.5% (11/20)	50% (5/10)	53.4% (31/58)

ACL: anterior cruciate ligament, PCL: posterior cruciate ligament; MCL: medial collateral ligament.

**Table 2 jcm-09-00973-t002:** Criteria for clinical assessment.

Clinical Parameter	Points	Excellent	Good	Fair	Poor
Subjective					
Pain		5	4	2	0
None	6				
Occasional pain, needs no medication	5				
Stabbing pain	4				
Intense, activity-related	2				
Night pain, at rest	0				
Walking capacity		6	4	2	1
Normal	6				
Outdoors >1 h	4				
Outdoors >15 min	2				
Indoors only	1				
Wheelchair/bedridden	0				
Objective					
Extension		6	4	2	2
Normal	6				
<10° loss	4				
>10° loss	2				
Total range of motion		5	4	2	1
>140°	6				
>120°	5				
>90°	4				
>60°	2				
>30°	1				
0°	0				
Stability		5	4	2	2
Normal	6				
Abnormal in 20° flexion	5				
Instability in extension <10°	4				
Instability in extension >10°	2				
Total (minimum)		30–27	26–20	19–10	9–6

**Table 3 jcm-09-00973-t003:** Criteria for radiologic assessment.

Radiological Parameter	Points	Excellent	Good	Fair	Poor
Depression		6	4	2	0
None	6				
<6 mm	4				
6–10 mm	2				
>10 mm	0				
Condylar widening		6	4	2	0
None	6				
<6 mm	4				
6–10 mm	2				
>10 mm	0				
Angulation (valgus/varus)		6	4	2	0
Normal	6				
<10°	4				
10°–20°	2				
>20°	0				
Total (minimum)		18	17–12	11–6	5–0

**Table 4 jcm-09-00973-t004:** Results of clinical assessment in 58 patients.

Fracture Type	No. of Patients	Average Clinical Score	Excellent	Good	Fair	Poor	Satisfactory Results
Type II	19	22.6 ± 5.3	4 (21%)	10 (52.6%)	5 (26.3%)	None	73.7%
Type III	2	26.0 ± 0.0	None	2 (100%)	None	None	100%
Type IV	7	25.3 ± 1.4	1 (14.3%)	6 (85.7%)	None	None	100%
Type V	20	23.6 ± 4.0	1 (5%)	17 (85%)	1 (5%)	1(5%)	90%
Type VI	10	21.6 ± 5.2	1 (10%)	5 (50%)	4 (40%)	None	60%
Total Injuries	58	23.2 ± 4.5	7(12%)	40(68.9%)	10(17.2%)	1(1.7%)	47(81%)

**Table 5 jcm-09-00973-t005:** Results of radiological assessment in 25 patients.

Fracture Type	No. of Patients	Average Radiological Score	Excellent	Good	Fair	Poor	Satisfactory Results
Type IV	5	16 (range: 14–18)	3 (60%)	2 (40%)	None	None	100%
Type V	2	16.5 (range: 14–18)	1 (50%)	1 (50%)	None	None	100%
Type VI	18	15 (range: 10–18)	9 (34%)	8 (44%)	1 (22%)	None	94%
Total Injuries	25	15.8	13 (52%)	11 (44%)	1 (4%)	None	96% (24/25)

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
