# Peer review of "Arthroscopy-Assisted Corrective Osteotomy, Reduction, Internal Fixation and Strut Allograft Augmentation for Tibial Plateau Malunion or Nonunion"

_jcm, 2020, doi:10.3390/jcm9040973_

Round 1

Reviewer 1 Report

Thank you very much for the opportunity to review the submitted manuscript "Arthroscopy-Assisted Corrective Osteotomy,Reduction,Internal fixation and Strut Allograft augmentation for Tibial Plateau Malunion or Nonunion".

The purpose of the current study was to present the results of Arthroscopy-Assisted Corrective Osteotomy (AACO), reduction, internal fixation and strut allograft augmentation for tibial plateau malunion or nonunion.

The analysis included fifty-eight patients, mean age 49±11.9 years , with tibial plateau malunion (n=44) or nonunion (n=14) into this study.

The mean follow-up period was 46.2 ±17.6 months. 

The manuscript presents new and interesting results, and the number of included cases is a major strength. Though I have comments and suggestions that may improve the quality and readability of the paper. Please see my below comments.

The articular depression was measured from the anteroposterior and lateral radiographs. In fact, measuring articular congruency in radiographs is possible, but inferior to computed tomography, especially in focal step-offs. Please comment. 

"Secondary osteoarthritis was diagnosed when radiographs showed a narrowed joint space in the injured knee at follow-up." What about clinical symptoms, where they considered as well? Please specify.

INTRODUCTION

Page 1, line 27: "Tibial plateau fracture is challenging for orthopedic surgeons because of the severity of trauma, 27 and associated soft tissue injuries." - You mean the management of tibial plateau fractureS?

Page 1, line 32: Traditionally, open corrective 32 osteotomy with bone grafting is standard of care following malunion [5,7-12]. - style ok. 

Page 1, line 33ff: However, arthrotomy 33 with wide soft tissue dissection may have the higher incidence of complications and longer recovery 34 time for the patient.[5,7,8,12] - style not ok!

Page 1, line 39: … removal of loose fragments[13-19].

Page 1, line 41: "previously.[13,14,20-23]"

MATERIALS AND METHODS

Please comment on Ethics declaration. 

All 58 patients completed a 139 questionnaire regarding their overall function. Was a Score system like Oxford Knee Score or Tegner Activity Scale used?

> The manuscript requires complete workover in terms of style / editing and language. 

RESULTS

page 8, line 215: "1 PCL complete rupture ,3 posterior cruciate ligament partial..."

line 216: "There were 4combined ligament injuries..." 

> please re-edit entire manuscript

DISCUSSION

page 9, line 247: "Huang et al analyze of the causes of failure for their 25 patients who had failed 247 tibia plateau surgery"

> please validate language by proof-reading service

page 10, line 287: "There were 51.7% patients suffered Schakter type V" 

page 10, line 294: "documemtation"

page 10, line 301: "determine further 300 osteoarthritic change needs to be longer" > full-stop missing!

> please re-read entire manuscript and improve language / definitions

What is the mean conversion time to TKA after ORIF for tibial plateau fractures? Please specify in your discussion and relate to your follow-up interval. 

Author Response

Pleeas see the attachment. 

Reviewer 2 Report

All the comments are in the attached file.

Reviewer 3 Report

This study is an article about the favorable results of arthroscopy-assisted corrective osteotomy in the malunion case of Tibial plateau fx.
However, some serious concerns are found in this article as follows.

First, as the authors suggest, early tibia plateau surgery appears to be fundamentally misguided. The surgeries seem to have been performed in ways that do not fit the general AO principle. 

Secondly, judging joint depression as simpla AP / lat radiographs in outcome measure ... this is very dangerous. Articular depression may be better judged by CT.

Third, the timing of evaluation of ROM after surgery is unclear, and there is no clear criteria such as evaluation criteria of the ROM or FC / FF.

Fourth, in line 95, the authors described 'the previous measured bone defect' ... How can you predict the extent of bone defects before surgery in a malunion case?

Fifth, please attach the intra-operative image intencifier (C-arm) photo in Figure 1 for our reader's understanding.

Sixth, as pointed out earlier, the cases shown in Figures 2 and 3 basically do not make sense for the initial OR / IF operation of tibial plateau fracture.

Seventh, it would be good to make a table of intraclass corelation coefficiency (ICC) for radiologic measurements.

Finally, as mentioned in the Journal ’s Instruction for Authors, you should use the PubMed abbreviations of Journal names.

Round 2

Reviewer 3 Report

Thank you very much for the authors' responses to our review.

*Each line refers to the revised manuscript.

Line 28-30: This paragraph seems too boring and cliché. Please delete it.

Line 51: Prospective study?? Very confusing. I think the design of this study is a retrospective case series.

Line 58-59: Please, add the reference regarding the Schatzker classification.

Line 64: Bad compliance? This is a very subjective item ... is there an objective criterion for this?

Line 74-75 (Table 1): As the authors answered in 'Response to reviewers’ comments', it is said that the ROM was measured for each OPD visit. In Table 1, it should be clearly indicated.

In Table 1, is the criteria for chondral injury Outerbridge classification? Please mark this to help the authors understand.

Line 88: do wound culture routinely? Is there any special reason or reference for this?

Line 111: ‘with two parallel K-wires’

Line 110-116 (Figure 2): The locked plate does not appear to be used in the postoperative x-ray in Figure 2. We don't even understand this. If it is a reoperation and articular depression is the main concern, shouldn't it have been performed with a locked plate?

Line 131: ACL injuries? Very confusing. Despite the reoperation due to malunion, what does ACL injury mean? Are you saying you were a reinjured patient?

Line 136: radiographic evidence of healing? Is there an objective or reference for Union? In general, most articles related to fractures usually describe this correctly or cite the references.

Line 136-138: For brevity and conciseness of the article, please delete this sentence. Just cite the reference.

Line 173: over a 1-year period? Of course, even if there is a difference between the articles ... Isn't that usually 6 months?

Line 184-191: As mentioned in the previous review, it would be better to describe the statistical method of ICC. Moreover, what kind of statistical program was used?

Finally, as mentioned in the previous review, you should use the PubMed abbreviations of Journal names.

Author Response

Please see attached filed. Thank you very much.
